# Towards Meaningful Interpretation of Molecular Data: Insights Gained from HMMD Challenges in *Salmonella* Detection for Future NGS Integration in Clinical Microbiology

**DOI:** 10.3390/diagnostics15010077

**Published:** 2024-12-31

**Authors:** Hyunji Kim, Soo Hyun Seo, Jae-Seok Kim, Kwang Jun Lee, Kyoung Un Park

**Affiliations:** 1Department of Laboratory Medicine, Seoul National University Bundang Hospital and Seoul National University College of Medicine, Seoul 03080, Republic of Korea; hyunji8713@snubh.org (H.K.); shseo@snubh.org (S.H.S.); 2Department of Infection Control Unit, Kang Dong Sacred Heart Hospital, Seoul 05355, Republic of Korea; jaeseokcp@gmail.com; 3Division of Zoonotic and Vector Borne Diseases Research, Center for Infectious Diseases Research, National Institute of Infectious Diseases, National Institute of Health, Cheongju 28159, Republic of Korea; kwangjun@korea.kr

**Keywords:** highly multiplexed microbiological/medical countermeasure diagnostic devices (HMMDs), next-generation sequencing (NGS), clinical utility, reflex culture testing

## Abstract

**Background:** With advancements in molecular diagnostics, including Highly Multiplexed Microbiological/Medical Countermeasure Diagnostic Devices (HMMDs) and the impending integration of Next-Generation Sequencing (NGS) into clinical microbiology, interpreting the flood of nucleic acid data in a clinically meaningful way has become a crucial challenge. This study focuses on the Luminex xTAG Gastrointestinal Pathogen Panel (GPP) for *Salmonella* detection, evaluating the impact of MFI threshold adjustments on diagnostic accuracy and exploring the need for an “indeterminate” result category to enhance clinical utility in molecular diagnostics. **Methods:** A retrospective review of *Salmonella*-positive cases detected via the Luminex xTAG GPP was conducted from June 2016 to November 2023. Key metrics included patient symptoms, stool culture results, and potential infection sources. Results were analyzed using the assay’s MFI cutoffs in Versions 1.11 and 1.12. Statistical comparisons between culture-confirmed and non-confirmed cases were performed using Kruskal–Wallis tests to assess MFI value distributions. **Results:** Among 2573 tests, 212 were *Salmonella*-positive under Version 1.11, while 185 were positive under Version 1.12. Adjusting the MFI threshold in Version 1.12 reduced false positives from 40.6% to 38.4% but led to one culture-confirmed positive case being missed. Statistically significant MFI differences were observed between culture-positive and culture-negative cases, suggesting that fixed binary cutoffs may not always yield clinically accurate interpretations. **Discussion:** The MFI threshold adjustment decreased false positives without fundamentally improving diagnostic accuracy, highlighting the limitations of binary interpretations in HMMDs. Introducing an “indeterminate” category, especially for cases with low MFI values, could aid clinicians in integrating molecular results with patient context. This approach offers a framework for future NGS integration, where nuanced interpretation will be essential to differentiate clinically significant findings from incidental data. **Conclusions:** Implementing an “indeterminate” interpretation category for HMMDs could enhance clinical decision-making and refine public health surveillance by focusing on clinically relevant findings. As NGS moves toward clinical application, establishing similar interpretive standards will be essential to manage the complexity and volume of molecular data effectively.

## 1. Introduction

Since the 1960s, microbiology has undergone significant evolution, with each decade bringing notable advancements. From the late 1990s to the early 2000s, molecular diagnostic technologies such as polymerase chain reaction (PCR) were widely implemented in the medical field. The introduction of Nucleic Acid Amplification Tests (NAATs) in microbial diagnostics addressed the limitations of traditional, time-consuming culture methods, enabling more rapid and specific detection of pathogens. This development led to the emergence of culture-independent diagnostic tests (CIDTs), specifically designed to overcome the challenges associated with traditional culture techniques [1].

The early 2000s marked a pivotal shift with the advent of molecular methods, which facilitated the development of CIDTs capable of detecting multiple pathogens from a single specimen, thereby bypassing the need for culture. By the late 2000s and into the 2010s, the incidence of infections involving multiple pathogens increased, creating a demand for diagnostic technologies capable of simultaneously detecting a broad spectrum of pathogens for public health and biodefense purposes. This need drove the development of Highly Multiplexed Microbiological/Medical Countermeasure Diagnostic Devices (HMMDs) [2]. Building on CIDT technology, HMMDs introduced advanced multiplex approaches, allowing for the simultaneous diagnosis of multiple pathogens (Figure 1).

In January 2013, four HMMDs for detecting acute gastroenteritis (AGE) pathogens received approval from the Food and Drug Administration (FDA), resulting in their rapid integration into microbiology laboratories. These devices represented a significant advancement in the field, meeting the growing need for comprehensive and efficient pathogen detection in clinical settings [1,3,4,5].

Today, several FDA-approved multiplex nucleic acid-based assays are available for detecting foodborne pathogens associated with AGE. These assays include the BIOFIRE FILMARRAY Gastrointestinal Panel (bioMérieux, Marcy-l’Étoile, France), xTAG Gastrointestinal Pathogen Panel (Diasorin, Saluggia, Italy), BD MAX Enteric Bacterial Panel (Becton Dickinson, Franklin Lakes, NJ, USA), QIAstat-Dx Gastrointestinal Panel 2 (QIAGEN, Hilden, Germany), and Biocode Gastrointestinal Pathogen Panel (Applied BioCode, Santa Fe Springs, CA, USA). These tests are capable of simultaneously detecting between 9 and 15 pathogens, including bacteria, viruses, and parasites, within approximately 5 h (Table 1, refer to Appendix A for more detailed information). According to the Centers for Disease Control and Prevention (CDC), the use of HMMDs for AGE pathogen detection has surged over the past decade. Healthcare providers are increasingly adopting these tests and DNA-based syndromic panels due to their rapid results and user-friendly nature compared to conventional culture methods [6]. By 2023, HMMDs were used in 78% of bacterial infection diagnoses, with 46% relying solely on these tests [7].

HMMDs offer several advantages, including faster turnaround times for targeted treatment, the ability to detect or rule out multiple pathogens from a single test, and potentially higher sensitivity compared to traditional culture methods. They also hold promise in resource-limited settings. However, HMMDs face several challenges, such as the uncertain clinical significance of some targets (e.g., Enteropathogenic *Escherichia coli*), the potential for multiple positive analytes in a single specimen, and the inability to distinguish between viable and non-viable cells. Additionally, HMMDs lack susceptibility information, may render specimens incompatible with culture-based tests, and do not provide culture results, which complicates public health efforts to obtain comprehensive infection-related data [8]. The sensitivity and specificity of HMMDs are critical factors for their effective public health application [9]. Therefore, the CDC recommends reflex culturing of specimens with positive HMMD results to enhance infection surveillance [10].

The first FDA-approved HMMD, the Luminex xTAG Gastrointestinal Pathogen Panel (GPP), employs nucleic acid amplification technology to simultaneously detect multiple targets from a single patient specimen. This technology combines PCR with bead-based multiplexing. Each target is assessed using specific mean fluorescence intensity (MFI) thresholds, although the rationale for these thresholds has not been extensively documented. For *Salmonella* detection, the Luminex xTAG GPP uses two probes with defined MFI thresholds. Probe 1, the primary probe, indicates positivity with an MFI above 1400 and negativity with an MFI below 200, while intermediate values (200–1400) are evaluated using probe 2. Probe 2 is set with an MFI threshold of 200 for positivity and below 200 for negativity. On 12 November 2019, the thresholds for *Salmonella* detection were revised, with probe 1’s threshold updated to 300 for negative, 100,000 for positive, and 300–100,000 for reflex to probe 2. This update aimed to enhance the stringency of positive calls and reduce false positives based on customer feedback. However, the impact of this revision on actual *Salmonella* detection has not been thoroughly reported.

This study aims to investigate the limitations of binary molecular interpretations in clinical microbiology by evaluating the impact of changes in the MFI cutoff on *Salmonella*-positive results using the Luminex xTAG GPP. Beyond assessing these technical adjustments, the study proposes the introduction of an “indeterminate” category for borderline molecular results, addressing the diagnostic gaps inherent in the current binary approach. By reviewing patient symptoms and stool culture outcomes, comparing results before and after threshold revisions, and identifying key areas for improvement, this research seeks to enhance laboratory practices, improve public infection surveillance, and inform the integration of future Next-Generation Sequencing (NGS) applications in clinical microbiology.

## 2. Materials and Methods

A retrospective chart review was conducted on patients who tested positive for *Salmonella* using the xTAG Gastrointestinal Pathogen Panel (Diasorin), an HMMD, between June 2016 and November 2023. The analysis focused on the patients’ symptoms, diagnoses, whether a stool culture was performed, and their exposure to potential sources of infection. The *Salmonella* positivity was re-evaluated based on the cutoff criteria from Versions 1.11 and 1.12 of the xTAG GPP.

Statistical analyses were conducted using R Version 4.2 (R Institute, Vienna, Austria). Non-parametric comparisons were made using the Kruskal–Wallis test to ensure robust analysis of the data.

## 3. Results

Out of 2573 tests conducted during the study period, 212 were confirmed as *Salmonella*-positive based on the Version 1.11 criteria, while 185 were positive under the Version 1.12 criteria. A total of 27 cases were identified as discrepant, where the results were positive according to Version 1.11 but negative under Version 1.12. Reflex cultures were selectively performed at the discretion of clinicians, primarily in patients with a clinical suspicion of *Salmonella* infection.

In Version 1.11, the distribution of results was as follows: 49.1% (*n* = 104) were both HMMD-positive and culture-positive, 40.6% (*n* = 86) were HMMD-positive but culture-negative, and 10.3% (*n* = 22) were HMMD-positive only. In comparison, under Version 1.12, the results were 55.7% (*n* = 103), 38.4% (*n* = 71), and 5.9% (*n* = 11), respectively (Figure 2). While the adjustment of the *Salmonella* detection probe threshold in the HMMDs reduced the number of false positives, it also resulted in one culture-confirmed positive case (3.7%) being incorrectly reported as negative.

When examining the clinical symptoms, those who were culture-confirmed for *Salmonella* did not exhibit distinct characteristics of infection exposure or key symptoms of *Salmonella* enteritis, such as abdominal pain, diarrhea, and fever, in the groups where the culture was not positive (Figure 3).

For the 185 samples that tested positive for *Salmonella* in both Version 1.11 and Version 1.12 of the Luminex xTAG GPP, the mean and standard deviation of the MFI values for probe 1 and probe 2 were analyzed based on whether stool cultures were performed and their results. Statistically significant differences were observed between groups (Table 2). Specifically, for probe 1, which is crucial for *Salmonella* detection, the MFI values were 2264.96 ± 1273.77 for stool culture-positive cases, 1232.49 ± 860.18 for stool culture-negative cases, and 1247.09 ± 1209.37 for cases where stool culture was not performed (*p*-value < 0.005).

## 4. Discussion

With advancements in molecular diagnostics, including technologies like HMMDs and the anticipated development of NGS, microbiology has entered a new era of data-driven insights. HMMDs, such as the Luminex xTAG GPP, enable rapid, multiplex pathogen detection directly from patient samples, providing valuable information on potential infections. These technologies have demonstrated significant utility in reducing turnaround time and detecting co-infections, as evidenced by prior studies highlighting their effectiveness across various bacterial, viral, and parasitic pathogens, particularly those difficult to culture. However, as molecular techniques generate vast amounts of nucleic acid data, they bring forth interpretive challenges that require more nuanced frameworks to ensure clinically meaningful outcomes.

Our study sought to evaluate the impact of manufacturer-introduced changes to the *Salmonella* MFI cutoff in the Luminex xTAG GPP, illustrating these interpretive challenges. After adjusting the cutoff, the rate of false positives in HMMD-positive but culture-negative cases slightly decreased from 40.6% to 38.4%. However, this adjustment also resulted in one culture-confirmed positive case (3.7%) being incorrectly reported as negative. The patient in this false-negative case, a 6-year-old girl with acute diarrhea and hematochezia, had a probe 1 MFI value below the newly established cutoff of 300, leading to a negative interpretation. This example highlights a critical issue: while false-positive results may arise from residual nucleic acids or cross-reactivity inherent in molecular testing, false-negative results may stem from low bacterial load or limitations in assay sensitivity [11,12]. The cutoff adjustment appeared to slightly refine the positive/negative boundary, but it did not improve diagnostic accuracy. Instead, this modification reflects a horizontal shift in threshold criteria, emphasizing the limitations of binary interpretation in molecular diagnostics.

It is noteworthy that the Luminex xTAG GPP does not provide detailed guidance on the principles behind positive judgment or the rationale for probe settings, leaving laboratories dependent on manufacturer-defined thresholds. This strict adherence to binary interpretation restricts diagnostic flexibility and, as demonstrated in this study, can lead to both false-positive and false-negative results. Such cases emphasize the need for more refined interpretive strategies, including the potential introduction of an “indeterminate” category for borderline results, to bridge the diagnostic gaps in molecular testing.

While the broader utility of the Luminex xTAG GPP remains indisputable [13]—particularly in its ability to rapidly detect a wide range of pathogens—our study highlights the importance of reevaluating the interpretive frameworks applied to HMMDs. This need extends beyond the specific case of the Luminex xTAG GPP to the broader challenges posed by binary molecular interpretations in clinical microbiology. As the integration of NGS into diagnostic workflows becomes increasingly imminent, it is critical to develop thoughtful and adaptive approaches that enhance the clinical utility of molecular data.

### 4.1. Balancing Data Sensitivity with Clinical Relevance: Lessons from HMMDs and NGS

As molecular diagnostics grow more sophisticated, they bring heightened sensitivity that can detect a broad range of pathogens, often identifying low levels of DNA that may not correlate with active infection. The anticipated integration of NGS into clinical microbiology promises even deeper insights, capturing microbial profiles in high detail. However, interpreting such expansive data will require a framework that differentiates clinically relevant findings from incidental or low-utility results.

For technologies like HMMDs, incorporating an “indeterminate” category could address this need by designating cases with ambiguous MFI values as requiring further clinical context. In the Luminex xTAG GPP, introducing an indeterminate category would allow clinicians to consider molecular findings alongside patient symptoms and history, particularly for MFI values that do not clearly indicate active infection. This approach aligns with future challenges anticipated with NGS, where data richness necessitates interpretive flexibility to avoid over-diagnosis and to ensure molecular findings are clinically actionable.

### 4.2. Trends in Public Surveillance of Salmonella Infections Pre- and Post-HMMD Implementation

Since the adoption of HMMDs, including the Luminex xTAG GPP, we have observed a rise in reported *Salmonella* cases, which may partly reflect the increased sensitivity of these tests in detecting low-infectivity or asymptomatic cases. This trend, observed nationally in South Korea, suggests that the greater sensitivity of HMMDs could influence public health surveillance by over-representing incidental or low-level infections. For example, *Salmonella* cases rose sharply in 2017, following the Luminex xTAG GPP introduction at our hospital in late 2016. With *Salmonella* classified as a high-priority pathogen by the WHO and as a Class 4 notifiable disease in South Korea, hospitals are required to report cases to the Korea Disease Control and Prevention Agency (KDCA) within seven days (Figure 4) [14,15]. However, while typical *Salmonella* incidence is influenced by seasonal temperature increases between June and August, the consistent rise in KDCA-reported cases since 2017 appears to be associated with HMMD use, independent of temperature trends (Figure 5) [16,17]. Notably, studies have reported false positives with the Luminex xTAG GPP for *Salmonella*, indicating that part of this increase may be attributed to HMMD-detected results with uncertain clinical significance [18,19,20].

### 4.3. The FDA’s Total Product Life Cycle (TPLC) Approach with HMMDs

The FDA’s Center for Devices and Radiological Health (CDRH) implemented the Total Product Life Cycle (TPLC) approach in 2019, ensuring continuous oversight of medical devices from development through to post-market use [21,22]. Since then, gastrointestinal pathogen panels, including HMMD-based assays, have been under FDA post-market surveillance. A recent rise in error alarms reported through medical device surveillance systems, primarily related to false positives, suggests potential limitations in relying solely on HMMDs for infection surveillance (Figure 6) [23]. Given that HMMDs often lack confirmable isolates, and many laboratories depend entirely on HMMD results, public health surveillance based solely on molecular data may be insufficient. This study specifically evaluated *Salmonella* detection with the Luminex xTAG GPP; therefore, findings should not be generalized to other HMMDs without further evaluation.

### 4.4. Future Directions for HMMD Adoption

HMMDs currently produce binary results, leaving reflex culture decisions primarily to clinician judgment. Since the COVID-19 pandemic, molecular diagnostics have become increasingly relied upon in microbiology, but an over-dependence on molecular results alone may pose clinical challenges. Our findings reveal variability in MFI thresholds for *Salmonella*, suggesting that probe MFIs related to different infectivity doses could be interpreted more flexibly.

This study was conducted on a heterogeneous group of patients, including those with true salmonellosis, convalescents, probable *Salmonella* carriers, and individuals with other gastrointestinal diseases. While this variability does not meet the strict inclusion and exclusion criteria of formal clinical trials, it reflects real-world diagnostic settings, where patient populations are often diverse. By analyzing this group, we aimed to evaluate the practical challenges associated with HMMDs and binary molecular interpretations, rather than conducting a controlled clinical trial. This approach emphasizes the importance of developing diagnostic frameworks that can address the complexities and uncertainties of molecular diagnostics in clinical practice.

Continuous monitoring and adjustment of thresholds, such as with the Luminex xTAG GPP assay, alongside efforts like the FDA’s TPLC activities, are encouraging developments that enhance the acceptance and reliability of molecular diagnostics. To further improve result interpretation, establishing an “indeterminate” category for cases with low MFI values would enable clinicians to combine these results with patient-specific clinical contexts and consider confirmatory cultures where necessary.

As molecular diagnostics continue to advance and NGS integration looms, adopting indeterminate interpretation facilitates more meaningful utilization of diagnostic data. Such an approach would help avoid over-diagnosis and offer a practical model for future NGS data handling, supporting nuanced clinical decision-making and accurate public health reporting.

## Figures and Tables

**Figure 1 diagnostics-15-00077-f001:**
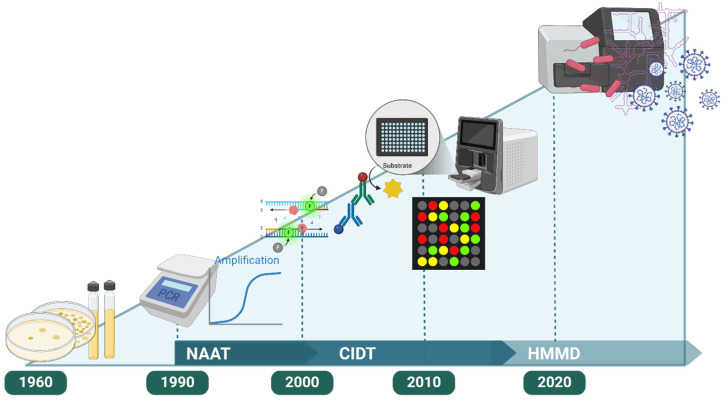
Advances in molecular diagnostics for infectious disease detection. The figure shows the progression from traditional culture methods to advanced molecular diagnostics, allowing for direct, multiplex detection of pathogens without the need for culture. With the introduction of PCR-based Nucleic Acid Amplification Tests (NAAT) in the 1990s, diagnostics evolved to include Culture-Independent Diagnostic Tests (CIDT) and array-based methods. This advancement continued into the 2020s with the development of Highly Multiplexed Microbiological/Medical Countermeasure Diagnostic Devices (HMMDs), enabling comprehensive multiplex technologies to simultaneously diagnose multiple pathogens.

**Figure 2 diagnostics-15-00077-f002:**
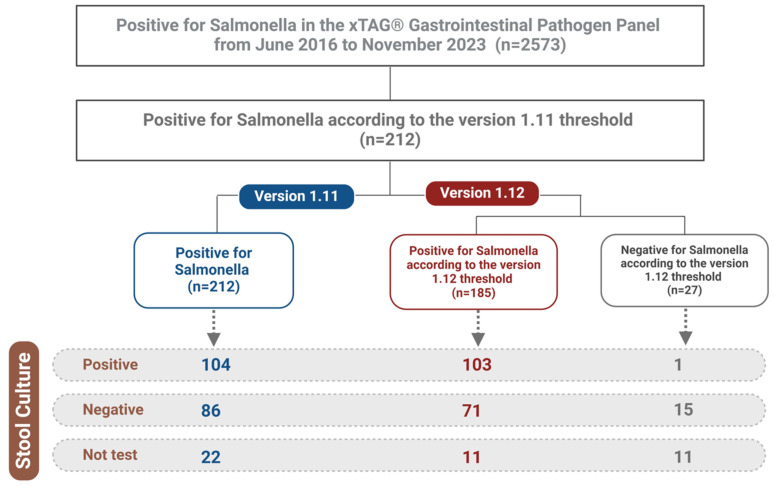
Overview of the study. Out of 2573 cases tested with the xTAG Gastrointestinal Pathogen Panel, 212 individuals were identified as *Salmonella* positive under Version 1.11 criteria. Following the revised threshold in Version 1.12, 27 of these patients were reclassified as negative.

**Figure 3 diagnostics-15-00077-f003:**
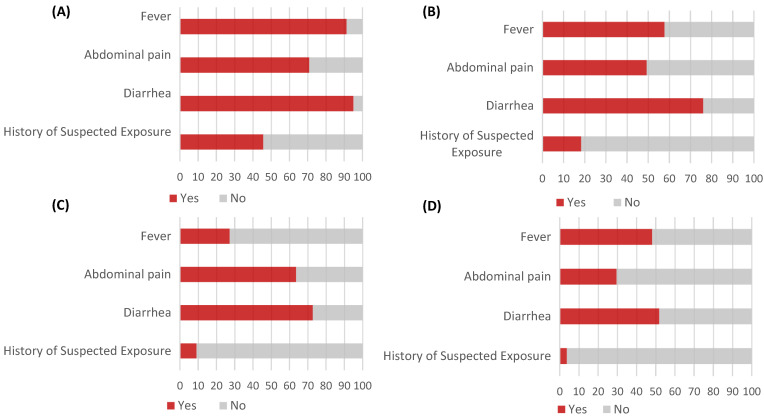
Clinical characteristics of patients positive for *Salmonella* on xTAG Gastrointestinal Pathogen Panel Version 1.11, classified by culture testing status and results. Excluding discrepant cases between Versions 1.11 and 1.12, patients were categorized as: (**A**) culture-positive, (**B**) culture-negative, (**C**) culture not performed, and (**D**) discrepant cases.

**Figure 4 diagnostics-15-00077-f004:**
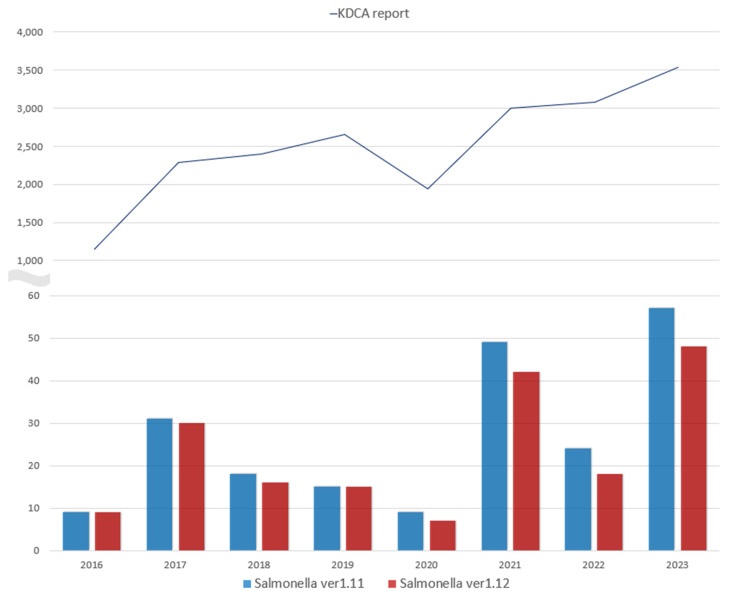
Annual reported cases of *Salmonella* to the KDCA and patients confirmed positive for *Salmonella* via the xTAG Gastrointestinal Pathogen Panel at our hospital. There was a sharp increase in reported cases of *Salmonella* to the KDCA in 2017, followed by a gradual increase in subsequent years. Similarly, SNUBH experienced a significant rise in the number of confirmed *Salmonella* cases in 2017, with a continued upward trend observed in 2021 and 2022.

**Figure 5 diagnostics-15-00077-f005:**
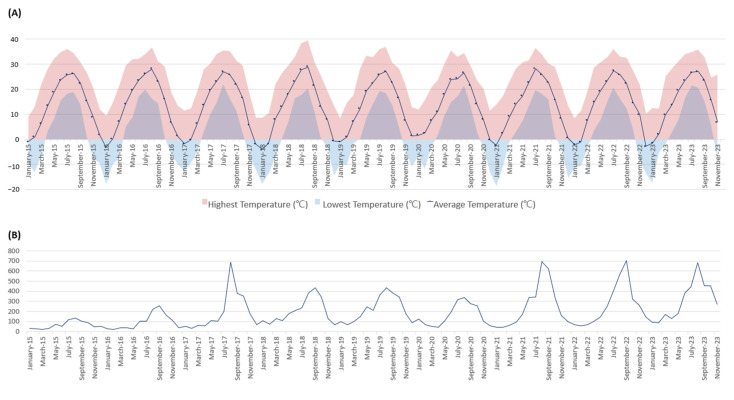
Temperature changes and reported cases of *Salmonella* to KDCA over the past nine years. (**A**) Monthly temperature changes from January 2015 to December 2023. There was no significant difference in average temperatures during the summer months from May to August. (**B**) Reported cases of *Salmonella* to the KDCA from January 2015 to December 2023. Due to the nature of *Salmonella* infections increasing with rising temperatures, there was an increase in reported cases during the summer months from May to August. Despite some fluctuations, there has been a general upward trend in the number of reported cases since 2017.

**Figure 6 diagnostics-15-00077-f006:**
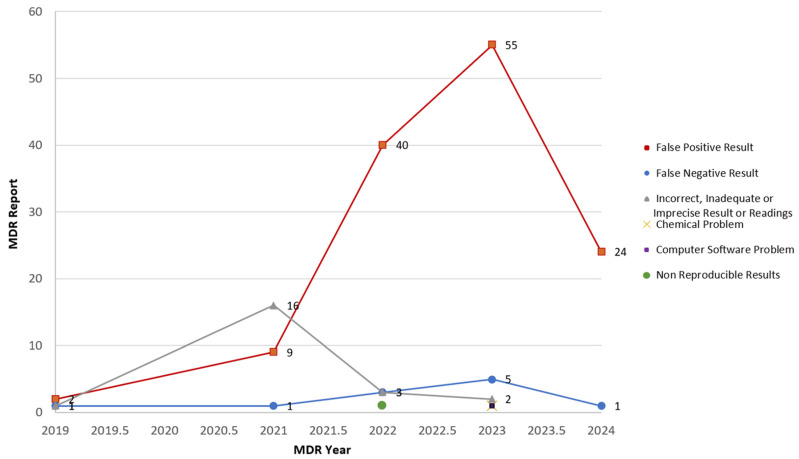
Number of MDR reports by year from June 2019 to April 2024. In September 2019, the FDA’s CDRH reorganized to focus on device oversight across the entire product lifecycle, from design and development to actual use, by implementing TPLC. Since its early implementation in 2019, post-market MDR data collected for the Gastrointestinal Pathogen Panel Multiplex Nucleic Acid-Based Assay System shows that false positive results have remained the most prevalent issue up to recent times.

**Table 1 diagnostics-15-00077-t001:** Comparison of FDA-approved Gastrointestinal Pathogen Panel Multiplex Nucleic Acid-Based Assay Systems.

Device Name	BIOFIRE FILMARRAY Gastrointestinal Panel	VERIGENE Enteric Pathogen Nucleic Acid Test	xTAG Gastrointestinal Pathogen Panel	BD MAX Extended Enteric Bacterial Panel	QIAstat-Dx Gastrointestinal Panel 2	Biocode Gastrointestinal Pathogen Panel
Company	BioFire Diagnostics, LLC	Nanosphere, Inc.	Luminex Molecular Diagnostics, Inc.	Becton, Dickinson and Company	QIAGEN Gmbh	Applied Biocode, Inc.
The First FDA Approval Date	2 May 2014	20 June 2014	21 March 2013	2 May 2017	31 May 2024	28 September 2018
Time to Result	~1 h	~2 h	5 h	~3 h	~1 h	<5 h
Number of Detected Targets	22 (13 bacteria, 5 viruses, and 4 parasites)	9 (5 bacteria, 2 viruses, and 2 toxins)	15 (9 bacteria, 3 viruses, and 3 parasites)	6 (4 bacteria and 2 toxins)	16 (8 bacteria, 4 viruses, and 4 parasites)	18 (12 bacteria, 3 viruses, and 3 parasites)

Abbreviations: FDA, Food and Drug Administration.

**Table 2 diagnostics-15-00077-t002:** Comparison of probe-specific MFI differences (mean ± standard deviation) based on stool culture results for *Salmonella*-positive samples, regardless of MFI threshold adjustments in the xTAG Gastrointestinal Pathogen Panel.

	Stool Culture-Positive	Stool Culture-Negative	Stool Culture Not Tested	*p*-Value
Probe 1	2264.96 ± 1273.77	1232.49 ± 860.18	1247.09 ± 1209.37	<0.005
Probe 2	2084.57 ± 1310.64	1521.54 ± 1040.57	1379.59 ± 1343.62	0.004

## Data Availability

The raw data supporting the conclusions of this article will be made available by the authors on request.

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
