# Peer review of "Towards Meaningful Interpretation of Molecular Data: Insights Gained from HMMD Challenges in Salmonella Detection for Future NGS Integration in Clinical Microbiology"

_diagnostics, 2024, doi:10.3390/diagnostics15010077_

Round 1
Reviewer 1 Report
Comments and Suggestions for Authors
The manuscript "Towards Meaningful Interpretation of Molecular Data: Insights from HMMDs for Future NGS Integration in Clinical Microbiology" has got a very intriguing title, which is unfortunately not supported with the manuscript (MS) body. This MS begins as review, but in the middle it becomes an experimental paper. Such combination is very strange, especially in context of hundreds papers published to date and devoted to evaluation of accuracy, sensitivity, specificity and other aspects of the xTAG® Gastrointestinal Pathogens Panel (GPP) test use in clinical practice.
The declared aim of the manuscript is not significant, it is a technical advance of the test manufacturer to improve the quality of diagnostics due to introduction of new cutoff criteria in v. 1.12 instead of v. 1.11 of the xTAG GPP software. The main result of the study is absolutely clear - improvement of salmonellosis diagnostics with new version of software. However, this result is neither new nor original in evaluation of laboratory tests generally. Besides, it is evident role of the test manufacturer to improve the software regularly to improve diagnostics. It is absolutely ordinary situation and cannot be the reason for clinical study.
The clinical study has got a great flaw, it does not contain two groups of patients, one with clear diagnosis, and another healthy group without salmonellosis confirmed with both cultural and serological methods. The study dwells on a group of patients which was a relly mixture of true salmonellosis patients, convalescents, probabe salmonella carriers, patients with other GI diseases. In such mixture it is impossible and useless to compare laboratory tests and evaluate their accurace, sensitivity, etc. Such study does not meet standards of clinical trials, honestly it cannot be considered as clinical trial at all, because it does not have criteria of inclusion and exclusion, exact description of compared groups, clear and significant purpose.
The main part of the manuscript is devoted to comparison between results of two lab methods of salmonellosis diagnostics, PCR/DNA hybridization xTAG® Gastrointestinal Pathogens Panel (GPP) kit and cultural method. However, all the body of papers published before and devoted to evaluation of xTAG® Gastrointestinal Pathogens Panel (GPP) kit diagnostic significance have been successfully ignored by authors, including recent metareview https://doi.org/10.1136/bmjgast-2020-000553. The readers have got false opinion that this MS is the first one comparing the xTAG® Gastrointestinal Pathogens Panel (GPP) test with cultural method.
Last part of the MS describing a sharp rise of salmonellosis in South Korea due to a wide application of the xTAG® Gastrointestinal Pathogens Panel (GPP) test seems very strange and contradicts trends in public health of developed countries. In addition, these data seem not to be true in light of the WHO recommendation to confirm all gastrointestinal infections with cultural tests as well gold stadard of salmonellosis diagnostics including mandatory cultural test. Moreover, the main and almost the only reason of increase of salmonellosis incidence is deterioration of sanitary conditions. Can the authors confirm it with epidemiological facts or appropriate conclusions of the South Korea public health service?
Thus, I have to note that the reviewed manuscript should be rejected due to very low levels of originality and significance, improper design, incorrect clinical trial, unclear and inexplicable conclusions.
Author Response
We sincerely appreciate your thoughtful feedback, which has allowed us to further clarify the intent and scope of our manuscript.
First, we would like to emphasize that this study is not intended to evaluate the technical accuracy or performance of the Luminex xTAG® Gastrointestinal Pathogens Panel (GPP). As you correctly noted, the introduction of new cutoff criteria in version 1.12 of the xTAG GPP software represents a technical advancement by the manufacturer aimed at enhancing diagnostic quality. We, too, acknowledge this as a positive and meaningful development in the field of molecular diagnostics.
That said, our study highlights a broader concern: the over-reliance on binary molecular diagnostic results ("positive" or "negative"), which may inadvertently lead to a reduced emphasis on confirmatory culture testing and clinical assessment. While molecular diagnostics provide rapid and sensitive detection, strict binary interpretations can sometimes fail to capture critical clinical nuances. For example, as many clinicians may have observed, adjustments to diagnostic thresholds—such as those implemented in the Luminex GPP software—can result in false negatives for culture-positive patients. This variability underscores the need for a more balanced approach that incorporates patient infection status and clinical context when interpreting results.
The central aim of our manuscript is to propose the introduction of an “indeterminate” category for borderline molecular results. We believe this approach can improve clinical decision-making by encouraging the integration of molecular findings with patient-specific clinical information and confirmatory culture testing where appropriate. While this perspective may differ from conventional experimental studies, we felt it was essential to base our discussion on practical evidence and align it with evolving recommendations from organizations such as the FDA and CDC.
With regard to the increase in reported salmonellosis cases in South Korea, we respectfully suggest that this trend cannot be explained solely by a deterioration in sanitary conditions. Instead, we propose that the rise coincides with the adoption of molecular diagnostics, which have significantly enhanced detection sensitivity. This shift in diagnostic practices likely accounts for the increase, as previously undetected cases are now being identified through advanced molecular tools—a phenomenon we describe as a diagnostic shift rather than a true epidemiological change.
We acknowledge that the structure of our manuscript may feel unconventional, as it combines both experimental insights and conceptual discussions. However, we believe this approach is necessary to provide a comprehensive perspective that fosters a more nuanced and clinically meaningful interpretation of molecular diagnostic results.
We are truly grateful for your detailed and constructive comments, which have helped us clarify our manuscript’s focus and strengthen its overall presentation.

Reviewer 2 Report
Comments and Suggestions for Authors
The study is quite interesting and is important to communicate the fine differences according to the advances of the programs, as well as the current limitations.
In general, culture methods are known to be less sensitive than molecular methods, and as the authors demonstrated, binary interpretations in HMMDs also have limitations. For me, it sounds difficult to implement an “indeterminate” interpretation category for HMMDs by focusing on clinically relevant findings. Maybe it could be possible in settings with a low incidence of parasitism.
Could you please modify Figure 6?. It is quite difficult to see the points of problems and non-reproducible results.
Author Response
Commnets: Could you please modify Figure 6?. It is quite difficult to see the points of problems and non-reproducible results.
Response: Thank you for your insightful feedback. We appreciate your comments regarding the limitations of culture methods compared to molecular methods and the challenges surrounding binary interpretations in HMMDs. We agree that introducing an “indeterminate” category could be challenging, particularly in clinical practice. However, as you suggested, this approach may indeed be more feasible in settings with a low incidence of parasitism, and we believe this direction warrants further exploration.
Regarding Figure 6, we have carefully revised the figure to improve clarity and ensure that points representing problems and non-reproducible results are more visible. Specifically:
The chemical problem that occurred in 2023 (1 case),
The computer software problem that also occurred in 2023 (1 case), and
The non-reproducible error observed in 2022 (1 case),
were previously less noticeable due to their size. To address this, we have increased the size of these data points for better visibility while maintaining the overall layout of the figure. We hope this revision enhances the readability and clarity of the figure.
Thank you again for your valuable suggestions, which have helped us improve the quality of our work.

Reviewer 3 Report
Comments and Suggestions for Authors
The article addresses the critical need for meaningful interpretation of molecular diagnostics in clinical microbiology, focusing on Highly Multiplexed Microbiological/Medical Countermeasure Diagnostic Devices (HMMDs) and Next-Generation Sequencing (NGS). Authors performed comprehensive analysis of Salmonella detection through retrospective data, using well-documented analytical tools. Recommendation to implementing an “indeterminate” category to improve diagnostic accuracy and clinical decision-making are thoughtful and actionable.
Well documented study with clear interpretation.
Few minor comments:
1. Table 1 is too complicated and hard to read. Make it simple. Remove information that are not relevant to the this manuscript and move it to supplement.
2. Increase font size in Figure 3, to make it easier for the reader.
Author Response
Comments1: Table 1 is too complicated and hard to read. Make it simple. Remove information that are not relevant to the this manuscript and move it to supplement.
Response: Thank you for your valuable feedback regarding Table 1. As suggested, we have simplified Table 1 by removing information that is not directly relevant to this manuscript. The removed content has been relocated to the supplementary materials for better readability and accessibility. We hope this improves the clarity and presentation of the table.
Comments2: Increase font size in Figure 3, to make it easier for the reader.
Response: Thank you for pointing this out. We have increased the font size in Figure 3 to enhance readability and ensure a more comfortable experience for the reader. We appreciate your suggestion, which has helped us improve the presentation of our figures.

Round 2
Reviewer 1 Report
Comments and Suggestions for Authors
The manuscript "Towards Meaningful Interpretation of Molecular Data: Insights from HMMDs for Future NGS Integration in Clinical Microbiology" has got a very intriguing title, which is unfortunately not supported with the manuscript (MS) body. The authors have corrected the MS. However, it remains unclear and has great inconsistencies, which prevent its publishing. I strongly recommend to improve the MS, because without additional explanation, which has been provided in the authors' reply, the manuscript looks unclear and incomplete.
First, the title of the MS looks like a title of the comprehensive review, not research article and should be changed to focus better on the drawbacks of the molecular methods and their insufficiency for confirmation of infectious disease, particularly salmonellosis while using solely.
Second, the aim of MS declared by the authors is "to assess the impact of changes in the MFI cutoff on Salmonella positive results using the Luminex xTAG® GPP." As I said before, The declared aim of the manuscript is not significant, it is a technical advance of the test manufacturer to improve the quality of diagnostics due to introduction of new cutoff criteria in v. 1.12 instead of v. 1.11 of the xTAG GPP software. directly However, in their answer the authors have indicated directly that their true purpose is "to propose the introduction of an “indeterminate” category for borderline molecular results". So, the authors should modify the aim to provide clear understanding of their intentions.
Third, the possible reasons of false-positive and false-negative results of both molecular and cultural methods and their discussion would provide a clear indication to the problem of “indeterminate” results of laboratory diagnostics. That discussion would be especially useful to provide possible causes of clinical symptoms revealed in the people with negative results of salmonella lab diagnostics.
Fourth, the clinical study has got a great limitation. The study dwells on a group of patients which was really a mixture of true salmonellosis patients, convalescents, probabe salmonella carriers, patients with other GI diseases. Such study does not meet standards of clinical trials, honestly it cannot be considered as clinical trial at all, because it does not have criteria of inclusion and exclusion, exact description of compared groups, clear and significant purpose. So, the authors should provide clear indication to this limitation and give a sufficient explanation, why they considere this heterogenous group acceptable for their purpose.
Fifth, the main part of the manuscript is devoted to comparison between results of two lab methods of salmonellosis diagnostics, PCR/DNA hybridization xTAG® Gastrointestinal Pathogens Panel (GPP) kit and cultural method. However, all the body of papers published before and devoted to evaluation of xTAG® Gastrointestinal Pathogens Panel (GPP) kit diagnostic significance have been successfully ignored by authors, including recent metareview (https://doi.org/10.1136/bmjgast-2020-000553). To avoid a false opinion of readers that this MS is the first one comparing the xTAG® Gastrointestinal Pathogens Panel (GPP) test with cultural method, suitable results should be mentioned, and correspondent references should be provided.
Author Response
Recommendation 1
The manuscript "Towards Meaningful Interpretation of Molecular Data: Insights from HMMDs for Future NGS Integration in Clinical Microbiology" has got a very intriguing title, which is unfortunately not supported with the manuscript (MS) body. The authors have corrected the MS. However, it remains unclear and has great inconsistencies, which prevent its publishing. I strongly recommend to improve the MS, because without additional explanation, which has been provided in the authors' reply, the manuscript looks unclear and incomplete.
1) First, the title of the MS looks like a title of the comprehensive review, not research article and should be changed to focus better on the drawbacks of the molecular methods and their insufficiency for confirmation of infectious disease, particularly salmonellosis while using solely.
- Response: Thank you for your valuable feedback on the manuscript title. We understand your concern that the original title may have given the impression of a comprehensive review rather than a research article. In response, we have revised the title to better align with the focus on the limitations of molecular methods and their insufficiency in confirming infectious diseases, particularly salmonellosis, as follows: "Towards Meaningful Interpretation of Molecular Data: Insights Gained from HMMD Challenges in Salmonella Detection for Future NGS Integration in Clinical Microbiology". Our study aims to highlight the issues that may arise when the current reliance on binary molecular test results (positive/not detected) in other specialized fields is directly applied to molecular microbiology without consideration. This research revisits the current status and underscores its implications, providing significant insights. While the original title aligned with this broader direction, we have refined it to explicitly focus on Salmonella, as demonstrated in our findings, in accordance with your suggestion. We sincerely appreciate your constructive feedback, which has helped us improve the clarity and focus of both the title and manuscript. We hope this revised title better reflects the content and purpose of our research.
2) Second, the aim of MS declared by the authors is "to assess the impact of changes in the MFI cutoff on Salmonella positive results using the Luminex xTAG® GPP." As I said before, The declared aim of the manuscript is not significant, it is a technical advance of the test manufacturer to improve the quality of diagnostics due to introduction of new cutoff criteria in v. 1.12 instead of v. 1.11 of the xTAG GPP software. directly However, in their answer the authors have indicated directly that their true purpose is "to propose the introduction of an “indeterminate” category for borderline molecular results". So, the authors should modify the aim to provide clear understanding of their intentions.
- Response: Thank you for your valuable feedback regarding the declared aim of the manuscript. We appreciate your insight that the previously stated aim might have appeared limited in scope, focusing primarily on the technical adjustments of the xTAG® GPP software. Based on your recommendation, we have refined the manuscript to ensure that the broader purpose of our study is more clearly articulated. The revised aim now reflects our intention to address the limitations of binary molecular interpretations in clinical microbiology and to propose the introduction of an “indeterminate” category for borderline molecular results. We have incorporated these aspects throughout the manuscript to better highlight the diagnostic gaps and implications of our findings for future advancements in molecular microbiology, including NGS integration. We are grateful for your constructive suggestions, which have helped us enhance the clarity and focus of the manuscript.
3) Third, the possible reasons of false-positive and false-negative results of both molecular and cultural methods and their discussion would provide a clear indication to the problem of “indeterminate” results of laboratory diagnostics. That discussion would be especially useful to provide possible causes of clinical symptoms revealed in the people with negative results of salmonella lab diagnostics.
- Response: Thank you for highlighting the importance of discussing the possible reasons behind false-positive and false-negative results. In the revised discussion, we have addressed the false-negative case identified in our study, describing the patient’s clinical symptoms and the molecular data that contributed to the incorrect interpretation. Additionally, we emphasize how the binary interpretive framework and the lack of an “indeterminate” category could lead to such diagnostic challenges. This revision provides a clearer link between the study findings and the broader implications for laboratory diagnostics.
4) Fourth, the clinical study has got a great limitation. The study dwells on a group of patients which was really a mixture of true salmonellosis patients, convalescents, probabe salmonella carriers, patients with other GI diseases. Such study does not meet standards of clinical trials, honestly it cannot be considered as clinical trial at all, because it does not have criteria of inclusion and exclusion, exact description of compared groups, clear and significant purpose. So, the authors should provide clear indication to this limitation and give a sufficient explanation, why they considere this heterogenous group acceptable for their purpose.
- Response: Thank you for your detailed feedback regarding the limitations of our study. We agree that the heterogeneity of the patient group, including true salmonellosis cases, convalescents, probable Salmonella carriers, and individuals with other gastrointestinal diseases, is a limitation that warrants acknowledgment. While this variability does not meet the rigorous inclusion and exclusion criteria typical of clinical trials, we believe it reflects the complexities of real-world diagnostic scenarios. This diverse patient population allowed us to explore the interpretive challenges associated with HMMDs in broader clinical contexts, which aligns with our study’s purpose of evaluating binary molecular interpretations and proposing a more nuanced diagnostic framework. In response to your comment, we have revised the discussion section to clearly acknowledge this limitation and provide a rationale for the inclusion of a heterogeneous group. We have emphasized that our approach focuses on highlighting the practical challenges in interpreting molecular diagnostic results, rather than attempting to conduct a controlled clinical trial. We sincerely appreciate your insightful comments, which have helped us refine our manuscript and ensure clarity in communicating the study’s scope and purpose.
5) Fifth, the main part of the manuscript is devoted to comparison between results of two lab methods of salmonellosis diagnostics, PCR/DNA hybridization xTAG® Gastrointestinal Pathogens Panel (GPP) kit and cultural method. However, all the body of papers published before and devoted to evaluation of xTAG® Gastrointestinal Pathogens Panel (GPP) kit diagnostic significance have been successfully ignored by authors, including recent metareview (https://doi.org/10.1136/bmjgast-2020-000553). To avoid a false opinion of readers that this MS is the first one comparing the xTAG® Gastrointestinal Pathogens Panel (GPP) test with cultural method, suitable results should be mentioned, and correspondent references should be provided.
- Response: We appreciate your suggestion to reference prior studies evaluating the diagnostic performance of the xTAG® Gastrointestinal Pathogens Panel (GPP) kit. We have now included the meta-review you suggested (https://doi.org/10.1136/bmjgast-2020-000553) to provide proper context for our findings. This ensures that readers understand the comparative aspect of our study in relation to existing literature and avoids any misinterpretation regarding its novelty. We sincerely thank you for your insightful comments, which have helped us improve the clarity and contextualization of our manuscript.

Round 3
Reviewer 1 Report
Comments and Suggestions for Authors
The manuscript "Towards Meaningful Interpretation of Molecular Data: Insights from HMMDs for Future NGS Integration in Clinical Microbiology" has been corrected by the authors. All unclarities and inconsistencies has been removed, and now nothing prevent its publishing.